# Effects of Thermally Oxidized Vegetable Oil on Growth Performance and Carcass Characteristics, Gut Morphology, Nutrients Utilization, Serum Cholesterol and Meat Fatty Acid Profile in Broilers

Ghulam Yaseen [1], Muhammad A. Sarfraz [1], Saima Naveed [1], Farooq Ahmad [2], Fehmeada Bibi [3], Irfan Irshad [4], Muhammad Asif [4], Talat N. Pasha [1,5] and Shafqat N. Qaisrani [1,*]

[1] Department of Animal Nutrition, University of Veterinary and Animal Sciences, Lahore 54000, Pakistan; ghulam.yaseen013@gmail.com (G.Y.); awais.sarfraz107@gmail.com (M.A.S.); saimamahad@uvas.edu.pk (S.N.); tnpasha@uvas.edu.pk (T.N.P.)
[2] Department of Zoology, The Islamia University of Bahawalpur, Bahawalpur 63100, Pakistan; farooq.ahmad@iub.edu.pk
[3] Department of Zoology, Multan Campus, University of Education Lahore, Multan 60700, Pakistan; fehmeada.bibi@ue.edu.pk
[4] Institute of Continuing Education & Extension, University of Veterinary and Animal Sciences, Lahore 54000, Pakistan; irfanirshad@uvas.edu.pk (I.I.); muhammad.asif@uvas.edu.pk (M.A.)
[5] Township Campus, University of Education Lahore, Lahore 54770, Pakistan
* Correspondence: shafqat.qaisrani@uvas.edu.pk

**Abstract:** The impacts of dietary levels of oxidized vegetable (sunflower) oil on growth performance, gut morphology, nutrients utilization, serum cholesterol and meat fatty acid profile were evaluated in Ross 308 straight-run (n = 192) day-old broilers. The broilers were arbitrarily distributed among four dietary treatments including; FVO: fresh vegetable oil (1 mEq kg$^{-1}$), LOO: low oxidized (20 mEq kg$^{-1}$), MOO: moderately oxidized (40 mEq kg$^{-1}$), and HOO: highly oxidized vegetable oil (60 mEq kg$^{-1}$) with 5% inclusion containing six replicates. Results revealed that the broilers consuming MOO and HOO based diets showed reduced ($p = 0.05$) feed intake, body weight gain and carcass weight accompanied by a poorer feed conversion ratio than those consuming FVO. Villus height, villus height to crypt depth ratio, ileal digestibility of crude protein ($p = 0.041$), crude fat ($p = 0.032$) and poly unsaturated fatty acids ($p = 0.001$) in thigh muscles were decreased, whereas crypt depth ($p = 0.001$), serum cholesterol levels ($p = 0.023$) and short chain fatty acids ($p = 0.001$) were increased ($p < 0.001$) by increasing dietary oxidation level. In conclusion, MOO and HOO exerted deleterious effects on growth, carcass weight, gut development and nutrients utilization. Low oxidized vegetable oil (20 mEq kg$^{-1}$), however, with minimum negative effects can be used as a cost effective energy source in poultry diets.

**Keywords:** oxidized vegetable oil; gut morphology; growth performance; fatty acid profile; broilers

## 1. Introduction

For optimal production performance, proper levels of energy along with appropriate dietary protein is a prerequisite in broiler diets [1]. Energy requirements cost about 70% of the total budget of poultry rations. Fats provide 2.25 times more energy than proteins and carbohydrates, and are composed of triacylglycerols, diacylglycerols, phospholipids and free fatty acids [2]. Besides the provision of essential fatty acids, fats are the vehicle for vitamins, alleviate heat stress [3], and improve diet palatability and absorption of fat-soluble vitamins [4]. Vegetable oils contain polyunsaturated fatty acids (PUFAs) with a better digestibility [5] and greater tendency of oxidation [6] and can be used up to 8% in poultry feed. The dietary inclusion level of oil, however, depends, beside other factors, on its peroxide value (POV) that is an indicator of its rancidity [7]. High risk of oxidation

due to prolonged contact with air, sunlight, metallic catalysts and poor storage conditions limits the use of oil in poultry feed [8]. Heating and deep frying of vegetable oils at various temperatures, likewise, also results in thermal oxidation reactions that lead to physical and chemical alterations [9] and the production of various lipid peroxidation products [10]. Peroxides are generated through a peroxidation process of unsaturated fatty acids leading to the production of secondary oxidation products including ketones, other low molecular weight acids [4] and malondialdehyde that are the markers for lipid peroxidation [11]. These products of oxidation react with lipids, fat-soluble vitamins and proteins in the diet thus reducing its nutrient content. Few of these oxidation products are toxic and have harmful effects on intestinal absorptive cells [12], resulting in decreased growth of broilers [5]. The influence of dietary oxidised vegetable oil on poultry production has been extensively reviewed [13]. Poultry rations supplemented with oxidized oil may cause oxidative stress [14], due to greater reactive oxygen species and may lead to poor growth performance in birds [15]. Dietary oxidized oil enhances the cholesterol and low-density lipoprotein levels, with a reduced serum immune globulin contents [16], digestive functioning [17], and poorer meat quality [18].

The influence of dietary oil oxidation on growth performance has been studied in broilers [19] and various aquatic animals [20,21]. There is, however, contrary data available about the use of oxidized vegetable oil in animal feed. Mazur-Kuśnirek et al. [15], for instance, observed no significant influence on growth performance of broilers fed oxidized (POV = 55.7 mEq kg$^{-1}$) oil-based diets. Hence, recovered oil from the restaurants or the food industry may be used in animal feed as a valuable energy source [22]. It has been reported that the addition of 1 to 2% oil reduces the dustiness of the feed, whereas 4 to 5% oil in the diet increases dietary energy for broiler chickens and young fattening turkeys. In contrast, consumption of oxidized fat resulted in a decline in feed palatability as well as intake [23], and finally poor growth in broilers [5].

The current trial was, therefore, performed to evaluate the impact of various levels of oxidized vegetable oil on zootechnical performance and carcass characteristics, gut morphology, nutrients utilization, serum cholesterol and meat fatty acid composition in broilers. It was hypothesized that used vegetable oil from food processing facilities and restaurants can replace the expensive fresh oil in broiler diets as an energy source without compromising their zootechnical performance.

## 2. Results

### 2.1. Growth Performance

The influence of oil oxidation on a broiler's growth performance is shown in Table 1. The oxidation levels of dietary vegetable oil significantly influenced growth performance (FI, BWG and FCR) during the overall experimental period. The broilers consuming LOO containing diets indicated no effect ($p > 0.05$) on FI, BWG and FCR during starter phase (0 to 21 days), whereas 4% reduced BWG and 5% poorer FCR in the grower phase (22 to 35 days) than those consuming the FVO containing diets. The dietary supplementation of MOO resulted in 7% decreased FI, 16% reduced BWG and 11% poorer FCR in the starter stage (0 to 21 days), and similarly, 7% reduced FI, 13% lower BWG and 7% poorer FCR in the grower period (22 to 35 days). The broilers consuming HOO diets exhibited 7% reduced FI, 15% lower BWG and 10% poorer FCR during the starter phase (0 to 21 days), whereas 3% reduced FI, 8% decreased BWG and 5% poorer FCR during the grower stage (22 to 35 days). Over the entire experimental period (0 to 35 days), broilers consuming MOO and HOO based diets indicated 7 and 9% reduction in FI, 14 and 16% decreased BWG accompanied by 7 and 8% poorer FCR, respectively, than those consuming FVO based rations, whereas no significant decrease in growth performance (FI and BWG) was detected in the group consuming LOO based diets compared with those fed on the control diet.

**Table 1.** Influence of various oxidation levels of vegetable oil on broiler's growth performance during starter, grower and overall period *.

| Effects | FI [1] | | | BWG [2] | | | FCR [3] | | |
|---|---|---|---|---|---|---|---|---|---|
| Age (Days) | 0–21 | 22–35 | 0–35 | 0–21 | 22–35 | 0–35 | 0–21 | 22–35 | 0–35 |
| Treatments ** | | | | | | | | | |
| FVO | 57.5 [a] | 135.5 [a] | 91.4 [a] | 46.7 [a] | 81.8 [a] | 62.7 [a] | 1.23 [b] | 1.66 [b] | 1.46 [c] |
| LOO | 58.2 [a] | 135.4 [a] | 89.1 [a] | 45.2 [a] | 78.5 [ab] | 59.5 [a] | 1.29 [b] | 1.72 [ab] | 1.50 [b] |
| MOO | 53.3 [b] | 126.5 [b] | 84.6 [b] | 39.3 [b] | 71.0 [c] | 54.0 [b] | 1.36 [a] | 1.78 [a] | 1.56 [a] |
| HOO | 53.7 [b] | 131.6 [ab] | 82.9 [b] | 39.7 [b] | 75.2 [bc] | 52.9 [b] | 1.35 [a] | 1.75 [ab] | 1.57 [a] |
| Pooled SE | 0.85 | 2.28 | 1.12 | 0.61 | 1.62 | 0.88 | 0.01 | 0.02 | 0.01 |
| *p*-value | 0.001 | 0.041 | 0.002 | 0.001 | 0.001 | 0.001 | 0.001 | 0.040 | 0.001 |

[a–c] Means in a column bearing dissimilar superscripts vary significantly ($p < 0.05$). * Each observation denotes the mean of 6 replicates (8 broilers per replicate). ** FVO = fresh vegetable oil (POV = 1 mEq kg$^{-1}$), LOO = low oxidized oil (POV = 20 mEq kg$^{-1}$), MOO = moderately oxidized oil (POV = 40 mEq kg$^{-1}$), HOO = highly oxidized oil (POV = 60 mEq kg$^{-1}$). [1] FI = feed intake. [2] BWG = body weight gain. [3] FCR = feed conversion ratio.

## 2.2. Slaughter Body Weight and Carcass Characteristics

The influence of various oxidation levels of vegetable oil on slaughter body weight (weight after removal of the inedible parts of the meat) and carcass characteristics in broilers is shown in Table 2. The results showed that the carcass characteristics including dressing and giblets percentages, breast part and leg quarter percentages remained unaffected ($p > 0.05$) with dietary supplementation of LOO, MOO and HOO. Carcass weight was, however, reduced by 5% in LOO, 11% in MOO and 11% in HOO based diet fed broilers than those consuming FVO based diet.

**Table 2.** Influence of different dietary oxidation levels of vegetable oil on carcass characteristics in broilers *.

| Treatments ** | Body Weight (g) | Slaughter Weight (g) | Dressing (%) [1] | Giblets (%) [1] | Breast Part (%) [1] | Leg Quarter (%) [1] |
|---|---|---|---|---|---|---|
| FVO | 2195 [a] | 1454 [a] | 66.2 | 4.46 | 27.3 | 19.2 |
| LOO | 2083 [ab] | 1375 [b] | 66.0 | 4.47 | 26.4 | 19.3 |
| MOO | 1890 [bc] | 1240 [c] | 65.6 | 4.56 | 25.7 | 19.8 |
| HOO | 1851 [c] | 1210 [c] | 65.4 | 4.34 | 26.5 | 19.6 |
| Pooled SE | 47.1 | 31.4 | 0.97 | 0.47 | 0.68 | 0.42 |
| *p*-value | 0.001 | 0.001 | 0.70 | 0.87 | 0.48 | 0.73 |

[a–c] Means in a column bearing different superscripts vary significantly ($p < 0.05$). * Each observation denotes the mean of 6 replicates (3 broilers per replicate). ** FVO = fresh vegetable oil (POV = 1 mEq kg$^{-1}$), LOO = low oxidized oil (POV = 20 mEq kg$^{-1}$), MOO = moderately oxidized oil (POV = 40 mEq kg$^{-1}$), HOO = highly oxidized oil (POV = 60 mEq kg$^{-1}$). [1] Dressing, breast part, leg quarter, giblets (gizzard, heart, liver) were demonstrated as percentage of slaughter body weight of broiler chickens at 35 days of age.

## 2.3. Gut Morphology

The effects of various oxidation levels of vegetable oil on gut morphology in broilers are summarized in Table 3. Thermally oxidized vegetable oil based diets in broilers led to a lower ($p = 0.001$) VH, deeper ($p = 0.001$) crypts and reduced ($p = 0.001$) villus height to crypt depth ratio (VCR) than those fed with FVO containing diets. The results indicated 24 and 29% lower VH in the broilers fed with MOO and HOO based diets, respectively, than those fed with the control diet. The CD was increased by 7, 17 and 25% in the broilers consuming LOO, MOO and HOO based diets, respectively, than those fed with FVO containing diets. The broilers consuming LOO, MOO and HOO containing diets resulted in 8, 35 and 43% reduction in VCR, respectively, than those fed on FVO based diets. The histological depictions of duodenal villus height of all the dietary treatments are shown in Figure 1.

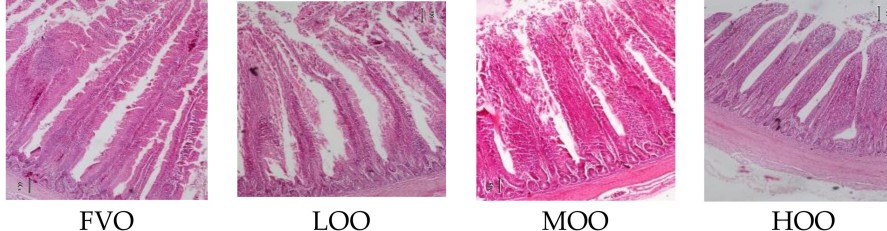

| FVO | LOO | MOO | HOO |

**Figure 1.** Duodenal villus height morphological representation of broilers fed various levels of oxidized vegetable oil. FVO = fresh vegetable oil (POV = 1 mEq kg$^{-1}$), LOO = low oxidized oil (POV = 20 mEq kg$^{-1}$), MOO = moderately oxidized oil (POV = 40 mEq kg$^{-1}$), HOO = highly oxidized oil (POV = 60 mEq kg$^{-1}$).

**Table 3.** Effects of oxidized vegetable oil on villus height (μm), crypt depth (μm) and villus height to crypt depth ratio in broilers *.

| Treatments ** | VH [1] | CD [2] | VCR [3] |
|---|---|---|---|
| FVO | 1450 [a] | 162 [c] | 8.95 [a] |
| LOO | 1427 [a] | 174 [c] | 8.20 [b] |
| MOO | 1096 [b] | 189 [b] | 5.79 [c] |
| HOO | 1037 [b] | 203 [a] | 5.11 [d] |
| Pooled SE | 27.0 | 5.00 | 0.16 |
| *p*-value | 0.001 | 0.001 | 0.001 |

[a–d] Means in a column bearing different superscripts are significantly different ($p < 0.05$). * Each observation denotes the mean of 6 replicates (3 broilers per replicate). ** FVO = fresh vegetable oil (POV = 1 mEq kg$^{-1}$), LOO = low oxidized oil (POV = 20 mEq kg$^{-1}$), MOO = moderately oxidized oil (POV = 40 mEq kg$^{-1}$), HOO = highly oxidized oil (POV = 60 mEq kg$^{-1}$). [1] VH = villus height. [2] CD = crypt depth. [3] VCR = villus height to crypt depth ratio.

### 2.4. Nutrients Utilization

Influence of oxidized vegetable oil on crude protein (CP) and crude fat (EE) digestibility in broilers is indicated in Table 4. The level of dietary oxidised oil linearly decreased digestibility of EE ($p = 0.032$) and CP ($p = 0.041$). The broilers consuming LOO-, MOO- and HOO-based diets had 1.6, 7.9 and 11.7% lower CP and 4, 6 and 11% reduced EE digestibility, respectively, in comparison with those consuming FVO containing diets.

**Table 4.** Influence of oxidized vegetable oil on nutrients utilization of crude protein (CP), crude fat (EE) and serum cholesterol concentration in broilers *.

| Treatments ** | CP (%) | EE (%) | Serum Cholesterol (mg/dL) |
|---|---|---|---|
| FVO | 68.5 [a] | 69.4 [a] | 82.1 [b] |
| LOO | 67.4 [a] | 66.5 [b] | 84.0 [b] |
| MOO | 63.1 [b] | 65.2 [bc] | 88.1 [b] |
| HOO | 60.5 [c] | 61.6 [d] | 106.0 [a] |
| Pooled SE | 1.20 | 0.05 | 5.32 |
| *p*-value | 0.041 | 0.032 | 0.023 |

[a–d] Means in a column bearing different superscripts vary significantly ($p < 0.05$). * Each observation denotes the mean of 6 replicates (3 broilers per replicate). ** FVO = fresh vegetable oil (POV = 1 mEq kg$^{-1}$), LOO = low oxidized oil (POV = 20 mEq kg$^{-1}$), MOO = moderately oxidized oil (POV = 40 mEq kg$^{-1}$), HOO = highly oxidized oil (POV = 60 mEq kg$^{-1}$).

### 2.5. Serum Cholesterol

The effects of oxidized vegetable oil on the concentration of serum cholesterol in broilers are mentioned in Table 4. The supplementation of broilers diets with oxidized vegetable oil indicated higher ($p = 0.02$) serum cholesterol concentrations compared with FVO. The broilers fed with LOO, MOO and HOO based diets indicated 2, 7 and 29% increased serum cholesterol concentration, respectively, than those consuming FVO based diets.

## 2.6. Meat Fatty Acid Profile

Table 5 indicates the effects of oxidized vegetable oil on the meat fatty acid profile of broilers. The concentrations of SFA, myristic acid (C14:0), palmitic acid (C16:0) and stearic acid (C18:0) were increased ($p < 0.05$), whereas PUFA, linoleic acid (C18:2), linolenic acid (C18:3), arachidonic acid (C20:4) and docosahexaenoic acid (C22:6) concentrations were reduced ($p < 0.05$) with the supplementation of LOO, MOO and HOO in the broiler diets than those fed with the FVO diet. The broilers consuming LOO-containing diets resulted in 4, 5, 3 and 4% increased concentrations of SFA, C14:0, C16:0 and C18:0, whereas 5, 5, 5, 9 and 10% reduction in the concentrations of PUFA, C18:2, C18:3, C20:4 and C22:6, respectively, than those consuming FVO based diets. The supplementation of MOO in the broiler diets led to 13, 12, 15 and 8% increase in the concentrations of SFA, C14:0, C16:0 and C18:0, whereas 12, 11, 6, 20 and 24% decrease in the concentrations of PUFA, C18:2, C18:3, C20:4 and C22:6, respectively, compared with those fed FVO-based diet. The broilers fed with HOO based diets indicated 21, 16, 27 and 10% higher concentration of SFA, C14:0, C16:0 and C18:0, whereas 24, 24, 13, 23 and 32% lower concentrations of PUFA, C18:2, C18:3, C20:4 and C22:6, respectively, than those consuming FVO in the diets.

**Table 5.** Effects of oxidized vegetable oil on meat fatty acid profile of broilers *.

| Treatments ** | Fatty Acids Profile (% of Total Fatty Acids) | | | | | | | | | | | |
|---|---|---|---|---|---|---|---|---|---|---|---|---|
| | C14:0 | C16:0 | C16:1 | C18:0 | C18:1 | C18:2 | C18:3 | C20:4 | C22:6 | SFA [1] | MUFA [2] | PUFA [3] |
| FVO | 0.43 [b] | 15.2 [d] | 2.20 | 8.97 [c] | 28.5 | 41.1 [a] | 0.80 [a] | 3.73 [a] | 0.41 [a] | 24.6 [b] | 30.7 [b] | 46.0 [a] |
| LOO | 0.45 [b] | 15.7 [c] | 2.46 | 9.33 [b] | 28.5 | 39.2 [b] | 0.76 [ab] | 3.39 [ab] | 0.37 [a] | 25.5 [b] | 31.0 [ab] | 43.7 [b] |
| MOO | 0.48 [ab] | 17.5 [b] | 2.60 | 9.67 [a] | 28.6 | 36.5 [c] | 0.75 [b] | 3.00 [b] | 0.31 [b] | 27.7 [ab] | 31.1 [ab] | 40.6 [c] |
| HOO | 0.50 [a] | 19.3 [a] | 2.65 | 9.90 [a] | 28.7 | 31.2 [d] | 0.70 [c] | 2.86 [b] | 0.28 [b] | 29.7 [a] | 31.2 [a] | 35.0 [d] |
| Pooled SE | 0.02 | 0.17 | 0.11 | 0.12 | 0.32 | 0.23 | 0.01 | 0.10 | 0.02 | 0.19 | 0.04 | 0.05 |
| *p*-value | 0.04 | 0.001 | 0.16 | 0.008 | 0.82 | 0.001 | 0.03 | 0.005 | 0.001 | 0.001 | 0.03 | 0.001 |

[a–d] Means in a column bearing dissimilar superscripts vary significantly ($p < 0.05$). * Each observation denotes the mean of 6 replicates (3 broilers per replicate). ** FVO = fresh vegetable oil (POV = 1 mEq kg$^{-1}$), LOO = low oxidized oil (POV = 20 mEq kg$^{-1}$), MOO = moderately oxidized oil (POV = 40 mEq kg$^{-1}$), HOO = highly oxidized oil (POV = 60 mEq kg$^{-1}$). [1] SFA = saturated fatty acid. [2] MUFA = monounsaturated fatty acid. [3] PUFA = polyunsaturated fatty acid.

## 3. Discussion

The current trial was performed to evaluate the impact of dietary oxidized vegetable oil on the growth performance and carcass characteristics, digestibility of CP and fat, gut morphology, serum cholesterol, and meat fatty acid profile in broilers consuming either fresh or oxidized vegetable oil-containing diets. Growth performance, carcass characteristics, gut morphology, serum cholesterol, utilization of crude protein and fat, and meat fatty acid profile were evaluated as descriptive variables.

The observed poorer growth performance in broilers consuming MOO and HOO based diets during starter period (0 to 21 days), grower phase (22 to 35 days) and overall (0 to 35 days) course of the study is in consistent with the studies in broilers reported earlier [5,24]. Tan et al. [25] reported poorer FCR in broilers consuming oxidized fish oil (POV = 140 mEq kg$^{-1}$) based diet in comparison with those consuming a diet containing fresh fish oil during 0 to 14 days of age. Lindblom et al. [26], similarly, documented reduced growth performance in the broiler groups consuming oxidized palm (POV = 607.4 mEq kg$^{-1}$), soybean (POV = 616.2 mEq kg$^{-1}$) and flaxseed (POV = 128.2 mEq kg$^{-1}$) oil in their diets than those consuming fresh oil counterparts. The reduction in FI and growth performance of broilers consuming dietary oxidized oil may be linked with its lower palatability and digestibility [27]. Lipid oxidation generates different oxidation products including esters, ketones, polymerized oils and aldehydes, which may cause the production of rancid odors and flavors that are ultimately associated with the decreased palatability of the diet [23]. Such oxidation products present in oxidized oil may lead to a lower energy value and fat retention of the diet, ultimately causing harmful effects on body weight of the broilers [5]. These oxidative compounds, if consumed, may damage the nutrients, mainly unsaturated fatty acids and antioxidative status of vitamin E leading to a

greater oxidative stress. The lower concentration of PUFAs and tocopherols along with the fatty acid polymerization in the oxidized oil may cause reduced fatty acid digestibility [28] and ultimately poor growth performance. The reduced growth, additionally, may also be associated with the destruction of fat-soluble vitamins (A, D, E and K), amino acids and pigments by rancid fats [27]. Oxidized oils in several animal species resulted in a reduced FI, depressed growth, and introduced disease [19,29,30]. Depressed growth performance by the consumption of oxidized fats may be due to the higher proliferation of hepatic cells, additionally, greater epithelial cell turnover and immunoglobulins concentration in gastrointestinal tract of broilers and pigs [31].

The decreased growth performance can also be attributed to reduce VH that exerts negative effects on the nutrients absorption by reducing the surface area available for digestive enzymes secretions in broilers [32]. This reduction in VH can also be due to the toxic properties of the oxidation products that cause destruction to the brush border membranes of the intestine. In the current study, the broilers consuming LOO based diets indicated no significant reduction in growth performance (FI and BWG) throughout the course of the experiment (0 to 35 days). Tan et al. [11] reported that the growth performance remained unaffected in broilers after consuming soybean oil of various oxidation levels (POV = 3.69, 25.37, 56.83 and 73.21 mEq $kg^{-1}$). This may possibly be due to the lower inclusion and oxidation levels of the oil in experimental diets. The decreased carcass weight in broilers consuming oxidized oil-containing diets is in accordance with the previous studies [24]. The current findings regarding dressing percentage, breast yield, giblet weights and leg quarter yield in broilers are compatible with the available literature on broilers [18,24,33,34].

The absorptive efficacy is mainly regulated by the villus surface area present for the nutrients. Improved VH equivalently enhances the surface area and consequently, higher intestinal digestive and absorptive functions [35]. The lower VH in broilers consuming oxidized vegetable oil is in accordance with reported literature on the poultry [7,36]. In contrast, Tan et al. [11] indicated no significant difference in the jejunal and ileal morphology of broilers that consumed oxidized (POV = 3.69, 25.37, 56.83 and 73.21 mEq $kg^{-1}$) oil-based diet. The reduced VH in the broilers fed with different oxidation levels of dietary vegetable oil, during the current study, may possibly be due to the toxic properties of some products of oxidation including ketones, esters, aldehydes and esters that are possibly causing damage to the intestinal brush border membrane [12]. This decrease in VH may also be due to an imbalance between the intestinal cell loss and regeneration rates [37]. Since dietary oxidized oil increases the proliferation rate and decreases the life span of the functional cells of gastrointestinal tract [38]. The broilers consuming oxidized oil may initiate intestinal metabolic oxidative stress [11].

The reduced EE digestibility in the present study is in accordance with Luo et al. [39], which indicated 8% reduction in the apparent total tract digestibility of EE in weaned pigs consuming 3% oxidized (POV = 120.85 mEq $kg^{-1}$) fish oil-based diets. Lindblom et al. [26], similarly, documented 6% reduction of EE digestibility in growing pigs consuming soybean oil, which was heated at 90°C for 72 hrs. Overholt et al. [40] reported 5% reduced EE digestibility in finishing pigs fed with the diet containing soybean oil that was heated at 180°C for 6 hrs. In contrast, no significant effects on EE digestibility by supplementation of oxidized oil (POV = 148 mEq $kg^{-1}$) in broilers [41] and nursery pigs were reported [42]. Lipid's digestibility depends on the numerous factors including the extent of saturation among others. Unsaturated fatty acids due to their greater capability of micelle formation, are usually more digestible than SFA. Oxidized oils containing lower unsaturated to saturated fatty acids ratio are, therefore, less digestible [43]. Lipid's oxidation increases the saturation of the fats [6,44], and the polymers generated by oxidation influence the digestibility of the lipids [45,46].

In the present study, the increased serum cholesterol level in the broilers consuming oxidized vegetable oil containing diets is in consistent with Açikgöz et al. [41], who found 5% increased concentration of cholesterol in broilers fed with the oxidized oil (POV =

148 mEq kg$^{-1}$) than the control group. Since the oxidation decreases the linoleic acid concentration in oil; therefore, the consumption of higher levels of dietary oxidised oil leads to a lower intake of linoleic acids, which may lead to an increase in total serum cholesterol level [47]. Ghasemi-Sadabadi et al. [7], similarly, reported 9% higher concentration of serum cholesterol in quails fed oxidized (POV = 146.03 mEq kg$^{-1}$) oil than those fed on the control diets. In contrast, Bayraktar et al. [48] reported 8% lower cholesterol concentration in plasma of broilers that consumed oxidized oil (POV = 100 mEq kg$^{-1}$) containing diets in comparison with those fed on the control diet. Yue et al. [30] also documented 4% lower cholesterol in serum of laying hens after consuming oxidized (POV = 88 mEq kg$^{-1}$) oil. This decreased cholesterol concentration may possibly be due to the reduced cholesterol uptake by liver, higher fecal excretion of cholesterol [49] and enhanced levels of thyroxin in the plasma [28].

The essential fatty acids composition of meat can be affected by certain fatty acids in the diet [50]. In the present study, a significantly lower percentage of PUFA, C18:2 and C18:3, and greater percentage of SFA and C18:0 in thighs of the broilers after consuming LOO, MOO and HOO based diets can be associated with heating of the oil. The heating or frying cause oxidation of the oil leading to decreased concentration of PUFA, C18:2 and C18:3 [1,19,51] that was reflected by the thigh tissue fatty acid profile [52]. In line with the current study, Ghasemi-Sadabadi et al. [7] documented a decreased concentration of PUFA, C18:2, C18:3 and C20:4 in the breast muscles of the quail that consumed oxidized (POV = 146.03 mEq kg$^{-1}$) oil. Mazur-Kuśnierek et al. [15], similarly, reported a lower PUFA concentration in broiler's breast muscles consuming oxidized oil (POV = 55.7 mEq kg$^{-1}$) than those fed the control diet.

## 4. Conclusions

In conclusion, supplementation of MOO (POV = 40 mEq kg$^{-1}$) and HOO (POV = 60 mEq kg$^{-1}$) exerted deleterious effects on growth, carcass weight, gut development, utilization of fat, and concentration of PUFA, C18:2 and C18:3 in the meat. Low oxidized vegetable oil (POV = 20 mEq kg$^{-1}$), however, with minimum negative effects, is a suitable energy source and can be added in poultry diets at 4 to 5% inclusion, replacing the FVO (POV = 1 mEq kg$^{-1}$) cost effectively.

## 5. Materials and Methods

### 5.1. Birds, Diets and Management

The trial procedure was revised and permitted by the animal ethics committee of University of Veterinary and Animal Sciences, Lahore, Pakistan. Ross (308) straight-run, day-old broilers (n = 192, 44.7 ± 0.8 g) were purchased from a commercial hatchery and placed in cage pens in an environmentally controlled facility. The broilers were randomly distributed over 24 cage pens to equalize initial weight between treatments and allocated to one of the four dietary treatments with six replicates each containing eight birds. Dietary treatments included FVO: fresh vegetable oil (POV = 1 mEq kg$^{-1}$), LOO: low oxidized vegetable oil (POV = 20 mEq kg$^{-1}$), MOO: moderately oxidized vegetable oil (POV = 40 mEq kg$^{-1}$), and HOO: highly oxidized vegetable oil (POV = 60 mEq kg$^{-1}$) to formulate corn-soy based diets with 5% inclusion for starter (0 to 21 days) and grower (22 to 35 days) periods (Table 6). The used vegetable (sunflower) oil was procured from a local food frying facility with POV of 20 mEq kg$^{-1}$ (LOO), which was further heated in a steel container with continuous stirring to attain the desired POV of 40 (MOO) and 60 (HOO) mEq kg$^{-1}$. Peroxide values were determined hourly by the American Oil Chemists Society [53] AOCS, 2007 Official Method Cd 8–53. Formulations were adapted by following the Ross nutrition specifications [54]. Experimental rations were provided to the birds in the form of mash throughout the study period (0 to 35 days). Each pen contained a separate nipple drinker and a tube feeder. *Ad libitum* supply of fresh water and feed was ensured for 24 h. A 23L:1D program was implemented for first 3 days and afterwards minimized to 16L:8D using 20 lux light intensity at bird level throughout the experiment. The shed

temperature was adjusted at 33 °C during the first 3 days and, later slowly decreased and maintained at 22 °C till the end of the study.

**Table 6.** Ingredients and nutrients composition of the experimental diets for broilers (as fed basis).

| Ingredients (%) | Starter (Day 0–21) | | Grower (Day 22–35) | |
|---|---|---|---|---|
| | Control | Treatment | Control | Treatment |
| Corn | 48.8 | 48.8 | 57.0 | 57.0 |
| Soybean meal (44%) | 35.4 | 35.4 | 30.6 | 30.6 |
| Wheat bran | 3.00 | 3.00 | 2.00 | 2.00 |
| Fresh vegetable (sunflower) oil * | 5.00 | - | 5.00 | - |
| Oxidized vegetable (sunflower) oil [1] | - | 5.00 | - | 5.00 |
| Canola meal | 4.00 | 4.00 | 2.00 | 2.00 |
| DCP [2] | 3.00 | 3.00 | 2.50 | 2.50 |
| Mineral premix [4] | 0.25 | 0.25 | 0.20 | 0.20 |
| Vitamin premix [3] | 0.20 | 0.20 | 0.20 | 0.20 |
| L-Lysine | 0.05 | 0.05 | 0.15 | 0.15 |
| DL-Methionine | 0.30 | 0.30 | 0.30 | 0.30 |
| Total | 100 | 100 | 100 | 100 |
| Calculated composition (%) | | | | |
| ME (kcal/kg) | 2988 | 2988 | 3100 | 3100 |
| DM | 88.9 | 88.9 | 88.6 | 88.6 |
| CP | 23.0 | 23.0 | 20.6 | 20.6 |
| EE | 2.85 | 2.85 | 2.94 | 2.94 |
| CF | 4.32 | 4.32 | 3.84 | 3.84 |
| Lysine-HCL | 1.26 | 1.26 | 1.10 | 1.10 |
| Methionine | 0.50 | 0.50 | 0.45 | 0.45 |
| Met + Cys | 0.91 | 0.91 | 0.84 | 0.84 |
| Arginine | 1.34 | 1.34 | 1.19 | 1.19 |
| Threonine | 0.83 | 0.83 | 0.73 | 0.73 |
| Valine | 0.93 | 0.93 | 0.84 | 0.84 |
| Analyzed composition (%) | | | | |
| ME (kcal/kg) | 2980 | 2978 | 3090 | 3087 |
| DM | 88.7 | 88.6 | 88.5 | 88.2 |
| CP | 22.8 | 22.8 | 20.5 | 20.5 |
| EE | 2.84 | 2.83 | 2.93 | 2.93 |
| CF | 4.30 | 4.31 | 3.83 | 3.83 |
| Lysine-HCL | 1.24 | 1.24 | 1.09 | 1.08 |
| Methionine | 0.49 | 0.49 | 0.44 | 0.44 |
| Met + Cys | 0.90 | 0.90 | 0.82 | 0.82 |
| Arginine | 1.33 | 1.32 | 1.17 | 1.17 |
| Threonine | 0.82 | 0.81 | 0.72 | 0.71 |
| Valine | 0.91 | 0.91 | 0.83 | 0.82 |

* Fresh vegetable oil with peroxide value = 1 mEq kg$^{-1}$. [1] Oxidized vegetable oil with peroxide values = 20, 40 and 60 mEq kg$^{-1}$ in both starter and grower diets. [2] DCP = dicalcium phosphate. [3] Vitamin premix (per kg of diet): vitamin B$_{12}$, 0.006 mg; riboflavin, 4.4 mg; pantothenic acid, 11 mg; choline chloride, 220 mg; folic acid, 0.55 mg; pyridoxine, 2.2 mg; biotin, 0.11 mg; thiamine, 2.2 mg; vitamin A, 5500 IU; vitamin D$_3$, 1100 IU; vitamin E, 11 IU. [4] Mineral premix supplied (mg per kg of diet): Manganese, 120; zinc, 100; iron, 60; copper, 10; iodine, 0.46; calcium, 150.

*5.2. Traits Measured*

5.2.1. Growth Performance and Carcass Characteristics

To determine the growth performance, FI, BWG, and FCR were determined at days 7, 14, 21, 28 and 35. The FCR was calculated per pen by dividing total FI by BWG together with the dead broiler(s). Mortality was, however, recorded every day. Feed intake was calculated by subtracting the leftover from the feed offered at the beginning of the week, whereas BWG was calculated by subtracting the final weight from the initial weight at the end of each week. On 35th day, three broilers per pen were arbitrarily chosen and slaughtered by the halal method. The dressing percentage (eviscerated carcass weight/slaughter body

weight) × 100, giblets (gizzard, heart and liver), leg quarter (drumstick with thigh) and breast weights (% of live weight) were evaluated.

### 5.2.2. Tissue Collection and Morphometric Examination

For morphometric analysis, the samples from the duodenum, where most of the digestion and absorption takes place, were collected from the slaughtered broilers as reported by Gopinger et al. [55]. From the middle of the duodenum, a sample of 2 cm length was obtained, dipped in cold normal saline (0.9% NaCl), preserved in a plastic container containing a 10% formalin solution. The samples were transferred to 70% ethanol in a period of 24 h, afterwards fixed and segmented at a thickness of 5 μm in paraffin. Six cross-sections from each broiler were processed using hematoxylin and eosin stain for histological study [56]. The VH (defined as distance starting the apex of the villus to the villus-crypt intersection) and CD (defined as the distance from the crypt-villus junction to the bottom) were determined on 10 undamaged, well-oriented villi (from collected specimen) from each broiler by utilizing the compound light microscope together with a camera. The images were analysed through Pixel Pro software.

### 5.2.3. Measurements of Ileal Digestibility of Crude Protein and Fat

*Ad libitum* supply of fresh water and feed was ensured to the broilers before slaughtering. After slaughtering and dissection, as mentioned before, ileal digesta was collected from three broilers per pen by gradually squeezing from Meckel's diverticulum to the ileocecal junction. This digesta was, afterwards, combined in a plastic container and stored at −20 °C for further study. The CP of the feed and the ileal digesta were determined by N × 6.25, where N was calculated using CuSO4 as a catalyst with Kjeldahl method (ISO 5983). Ileal digestibility of CP was measured using the equation below:

$$\text{IDCP (\%)} = \frac{\text{CPd/AiAd} - \text{CPi/AiAi}}{\text{CPd/AiAd}} \times 100$$

where; $CP_d$ = crude protein in the diet, $CP_i$ = crude protein in the ileal digesta, $AiA_d$ = acid insoluble ash in the diet, $AiA_i$ = acid insoluble ash in the ileal digesta.

Ileal digestibility of EE was calculated by the following formula:

$$\text{IDEE (\%)} = \frac{\text{EEd/AiAd} - \text{EEi/AiAi}}{\text{EEd/AiAd}} \times 100$$

where; $EE_d$ = crude fat in the diet, $EE_i$ = crude fat in the ileal digesta, $AiA_d$ = acid insoluble ash in the diet, $AiA_i$ = acid insoluble ash in the ileal digesta.

### 5.2.4. Blood Sample Collection and Serum Cholesterol Analysis

At the 35th day of age, three broilers per replicate (same broilers as for carcass traits) were arbitrarily chosen for blood sample collection before slaughtering. Blood samples (3 mL) were taken from the wing vein of each bird in yellow top, gel-coated vacutainer tubes. The blood samples were subsequently centrifuged at 1500× *g* for 10 min at 4 °C. Serum was separated and stored at −20 °C until further analysis. A Randox cholesterol, Colorimetric Method Kit (Randox Laboratories, Ardmore, Crumlin, UK) was used that is based on the CHOD-PAP method.

### 5.2.5. Meat Fatty Acid Profile Analysis

On the 35th day, thigh and drumsticks of three slaughtered broilers (the same broilers as for carcass traits) per pen, were collected and labelled for meat fatty acid profile. After collection, the meat samples were frozen (−20 °C) for further analysis. For fat extraction, the thigh and drumsticks were thawed, deboned and minced. The extraction of fat from the meat was performed using a cold solvent mixture following the methodology designated by [57] with chloroform, methanol and water as a solvent. The fatty acid methyl esters were prepared using the methodology given by [58]. The solvent of fatty acid methyl esters was

evaluated by GC-MS (7890-B, Agilent Technologies). Inlet and detector temperatures were set at 200 °C and 250 °C, with a 1:50 split ration using helium, hydrogen and oxygen at 1, 4 and 40 mL/min., respectively. The quantification was carried out according to FAME-37 standards [59].

### 5.3. Statistical Analysis

The data analysis was undertaken by the one-way ANOVA technique using GLM procedure (SAS software; version 9.1). For the measurement of significant differences between the means, a least significant difference test was applied. The following mathematical model was used for analysis:

$$Y_{ij} = \mu + ES_i + e_{ij}$$

where; $Y_{ij}$ = dependent variable, $\mu$ = mean of population, $ES_i$ = effect of energy source (oxidized or fresh vegetable oil), $e_{ij}$ = residual effect.

A probability level of 5% was used to describe significant differences.

**Author Contributions:** S.N.Q., G.Y., M.A.S., S.N., F.A., F.B. and I.I., equally contributed to conceptualization, data curation, formal analysis. T.N.P. and S.N.Q. were project administrators. Resources provided by S.N. and F.B., F.A., I.I. and M.A. contributed in software analysis. Original draft written by G.Y., M.A.S. and S.N.Q., S.N., F.A., F.B., I.I., M.A. and T.N.P. gave final review and editing. All authors have read and agreed to the published version of the manuscript.

**Funding:** This research received no external funding.

**Data Availability Statement:** The data of the published research is available, from the corresponding author, upon request.

**Conflicts of Interest:** The authors declare no conflict of interest.

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
