# Peer review of "Effects of Thermally Oxidized Vegetable Oil on Growth Performance and Carcass Characteristics, Gut Morphology, Nutrients Utilization, Serum Cholesterol and Meat Fatty Acid Profile in Broilers"

_catalysts, doi:10.3390/catal11121528_

Round 1

Reviewer 1 Report

Manuscript title/ Effects of Thermally Oxidized Vegetable Oil on Growth Performance and Carcass Characteristics, Gut Morphology, Nutrients Utilization, Serum Cholesterol and Meat Fatty Acid Profile in Broilers

  1. The whole manuscript need thoroughly English editing
  2. No line number was added
  3. All abbreviations in the abstract section should be mentioned for the first time
  4. The conclusion part of the abstract should be rewritten again
  5. Why the authors used the middle of the duodenum for Gut morphology studies
  6. The authors should be mentioned how to calculate FCR and other growth performance parameters
  7. The references in the introduction and discussion parts should be updated
  8. The paper design is well organized but, the Thermally Oxidized Vegetable Oil was used in many previous studies. I ask the authors to address the new topics in this manuscript
  9. As well as provide the figures of gut morphology
  10. Also, this study needs to examine the molecular pathway of Thermally Oxidized Vegetable Oil
  11. The authors showed that Cholesterol levels were 84, 88.1 for LOO and MOO respectively, can you explain.
  12. Why the authors did not assay the whole lipid profile, it is important

Author Response

Dear Reviewer, 

Please find attached our response regarding your valuable suggestion to improve the quality of the manuscript. 

Regards,

Dr. Shafqat

Reviewer 2 Report

The aim of the research was to okreÅ›lenie wpÅ‚ywu thermally oxidized sunflawer oil on growth performance and carcass characteristics, gut histology, nutrients utilization, serum cholesterol and meat fatty acid profile in chicken broilers.  The number of chickens used in the experiment is sufficient. The applied research methods are correct. The discussion is well conducted and comprehensive. Well-chosen references. Before publishing in Catalysts, the article requires additions and corrections. The proposed changes are listed below:

General comments:

Please prepare the article in accordance with the instructions for authors.

For affiliates, the first name and surname initials for each co-author of the article should be provided, the same as given in the "Author Contributions" chapter.

The missing chapter "Funding;  Acknowledgments;  Data Availability Statement, Conclusions" should be added

In the description of significance, use lowercase "p" in italics, spaces before and after "<" for example (p < 0.05)

In the References section, for a range of pages, use the long "-" from the insert function for all References items

Detailed comments:

Page 1,

Abstract chapter: L9 - "ileal digestability of crude protein and crude fat were decreased (P = 0.001) - inconsistent with data in table 5

L9 - "serum cholesterol ..." inconsistent with the data in Table 5

L13-14 "without ..." no compliance with the test results see FCR, PUFA, C18:2 content

Page 2

L5 FCR 5% poorer? or 4%

L8-9: Add something about the fat content of the compound feed mixtures, the addition of                  1-2% fat to reduce the dustiness of the feed, 4-5% fat to increase energy in diets for broiler chickens and young fattening turkeys

Page 3

Subchapter title 2.2. I propose to change to: Slaughetr body weight and carcass characteristics, and add description for slaughter BW

In table 3 Body weight (g) instead of Live weight. Dressing (%), Giblets (%), Breast part (%), Leg quarter (%) instead of current form

1Dressing, breast part, leg quarter, giblets (gizzard, heart, liver) were demonstrated as percentage of slaughter body weight of broiler chickens at 35 days of age - instead of current form

Page 4

No change

Page 5

Discussion chapter, L10

"Tan et al. [28] "instead of [28]

Page 6

L17 and 29 Tan et al. [12] instead of [12]

L39 "with Luo et al. [45]” instead of current form

Page 7

L2 "with Açikgöz et al. [47]” instead of current form

L4 "Ghasemi-Sadabadi et al. [42]” instead of current form

L6 „Bayraktar et al. [53]” instead of [53]

L8 „Yue et al. [34]” instead of [34]

L 19 "Mazur-KuÅ›nierek et al. [16]” instead of [16]

Page 9

L1-2 FI, BWG, FCR specified for 0 day?

L6 „(eviscetared carcass weight/slaughter body weight) x 100%, giblets” instead of current form

L10 „Gopinger et al. [61]” instead of [61]

L10 Why were the biometric features of duodenum not jejunum and ileum identified? This is where the main digestion and absorption takes place

Page 10

Add information about the verification of the normal distribution of traits, the name of the test used and the name of the test used to verify statistical differences between group means

Follow the Authors Contributions section as in the other articles in Catalysts journal

Page 12

Item 65, add an abbreviated name journal

Author Response

Dear Reviewer,

Please find attached our response regarding your valuable recommendations to improve the quality of the manuscript. We tried our level to do the needful.

Warm regards,

Dr. Shafqat

Round 2

Reviewer 1 Report

  1. why the authors select the middle of the duodenum for Gut morphology studies
  2.  the authors didn't add the FCR and other growth performance parameters calculation
  3. I ask the authors to address the new topics in this manuscriptbut the authors reply by (The reviewer’s suggestions can be opted in future studies because this study has already been carried out. At the moment no further changes regarding data collection are possible for this manuscript)?
  4.  The authors not provide the provide the figures of gut morphology?
  5. The authors showed that Cholesterol levels were 84, 88.1 for LOO and MOO respectively, can you explain, I asked again?
  6. lipid profile is important to this study
  7.  

Author Response

Dear Editor,

First of all, we would like to thank the Reviewers for their high quality and constructive suggestions and the Editor for your careful editing for our manuscript. All the comments and suggestions are valuable and very helpful for revising and improving our manuscript. We have made revisions according to the referees’ comments and suggestions as described in the authors’ response. In this revised version of the manuscript, we did our best to address all comments raised by the Reviewers.

Regards,
